# The Perceived Self-Efficacy of Teachers in the Use of Digital Tools during the COVID-19 Pandemic: A Comparative Study between Spain and the United States

**DOI:** 10.3390/bs13030213

**Published:** 2023-03-02

**Authors:** Judit García-Martín, Rodolfo Rico, Sheila García-Martín

**Affiliations:** 1Departamento de Psicología Evolutiva y de la Educación, Universidad de Salamanca, 37008 Salamanca, Spain; 2Colorado School of Mines, 1500 Illinois St, Golden, CO 80401, USA; 3Departamento de Didáctica General, Específicas y Teoría de la Educación, Universidad de León, 24071 León, Spain

**Keywords:** self-efficacy, teacher training, remote instruction

## Abstract

This study analyzed the use of fifteen groups of digital tools that 197 active teachers made during the sudden shift to remote instruction due to the COVID-19 quarantine orders placed by various health departments in Spain and the United States. The study also examined the impact that the use of digital tools had on teachers’ perceived self-efficacy. A quantitative research design was used, supported by an exploratory descriptive approach that materializes in the application of an online questionnaire during the spring of 2020. The results indicate that perceived self-efficacy differs from country of origin and is conditioned by sociodemographic variables such as the training received and type of center. It is noted that teachers in Spain prefer the use of Moodle or Escholarium over that of teachers in the United States that opted for Google Classroom as a primary platform for teaching online, and the frequency of use of digital tools analyzed does not guarantee that their implementation is effective.

## 1. Introduction

The outbreak of coronavirus that originated in China and became a pandemic in March of 2020 [1,2,3] spread exponentially through Spain and the United States, among other countries. With the goal of reducing the spread and mitigating the effects of the coronavirus, the World Health Organization (WHO) created a strategic plan and guiding documents for public safety and sanitation. In accordance with the aforementioned guidance from the WHO, most countries implemented social distancing measures and quarantine orders [2], forcing many learning institutions to suspend in-person teaching and focus their attention on virtual instruction, which required a shift from the usual teaching and learning situations at all education levels. In most cases, the shift to virtual instruction has materialized using various digital tools and occurs in virtual environments using videoconferences (synchronous sessions), while face-to-face classes also take place [4], which adds to the teacher’s workload.

Consequently, in March of 2020, teaching faculty were obligated to use their digital skills and technological capabilities to comply with the educational, social and health requirements during the COVID-19 pandemic and quarantine, suddenly becoming teachers 3.0, which ultimately means that the teaching process has been transformed and increases the prevailing need for teachers to be digitally literate [5].

## 2. Background

### The Technological Pedagogical Content Knowledge (TPACK) Model

One of the conceptual frameworks to measure the effective integration of digital tools in the educational field is Technological Pedagogical Content Knowledge (TPACK), which is an extension of the pedagogical content knowledge approach proposed by Shulman [6]. In relation to the TPACK diagram, in 2006, Mishra and Koehler [7] stated that for the integration of digital tools in the teaching of content to be effective, it is necessary for teachers to be trained in three dimensions of knowledge that are represented by three interlocking circles (see Figure 1). The first (CK) refers to the disciplinary content of the subject he teaches, the second (PK) to the pedagogical knowledge necessary to instruct students and for them to achieve the established learning objectives, and the third (CT) to the knowledge around the tools, in this case digital, that intervene in the teaching and learning process, enabling and optimizing the construction of meanings by students [7]. Studies also indicate that technological knowledge (TK) is essential for teachers’ confidence in content knowledge, pedagogical knowledge and effective technology integration [8]. Consequently, it is more than evident that the optimal integration of digital tools requires the interrelation of the three dimensions of knowledge [9]. This interrelation gives rise to four possible combinations: the PCK or pedagogical knowledge of the content that is materialized in the selection of the pertinent pedagogical elements depending on the subject that is taught; the TCK or knowledge of the use of available digital tools; the TPK or technological pedagogical knowledge; and the TPACK or technological, pedagogical, and content knowledge [7].

In accordance with this, previous studies have demonstrated that a principal factor in the effective use of these digital tools in educational settings is the attitudes and beliefs of self-efficacy of teachers [10] since, as evidenced by the TPACK model, attitudes alone are not enough for the successful integration of the tools in the teaching process for the effective delivery of instructional content but rather the beliefs that teachers have about the functionality of these tools is fundamental [11]. In this sense, prior research carried out thus far on the attitudes of teachers and their beliefs of self-efficacy demonstrates an interesting dichotomy: are they facilitators or limiting the integration of digital tools in the instructional process?

On one hand, previous studies have demonstrated that attitudes are an important predictor of the use of digital tools; for example, the one carried out by Tondeur et al. [12] in 2018, with 931 teachers in training from 20 teacher training institutions in Flanders. As are the beliefs of self-efficacy shown by the research carried out by Ertmer et al. [13] in 2012 and Farjon et al. [11] in 2019.

On the other hand, other recent studies have signaled that the attitudes and beliefs of teaching faculty on the use of digital tools are a barrier to the integration of the same tools and, due to the fear of change, the lack of training and personal use are presented as traditional obstacles to their integration in the teaching-learning process [13,14]. Research on emergency remote instruction also suggests that for effective technology integration, teachers require not only technology training but also strategies to cope with stressful situations [15].

Because of all the previously stated, self-efficacy, which refers to the belief that a person has about their ability to perform a behavior successfully [16], is presented as an important determinant of the behavioral intention of use, that is, there is reliable evidence that having a positive judgment about one’s ability influences the acceptance of digital tools in teaching [17]. In fact, a recent study reveals that pre-service teachers with higher levels of internet self-efficacy and lower levels of anxiety are more likely to have higher levels of digital citizenship [18].

## 3. This Study

Under this health, social and cultural panorama, the purpose of this study is to analyze the use that active teachers made of fifteen groups of digital tools during the months of confinement in Spain and the United States and the impact that this use had on their perceived self-efficacy.

All of this is to glimpse some possible lines of action that will make virtual learning more effective and efficient the next time. In this sense, there are few studies that address this issue in active teaching staff and, even fewer, those that do so considering a sample of the Spanish and American population at an exceptional moment. All the above gives rise to the following research question: what can teachers learn about what happened during this difficult time that will make virtual learning better next time?

### 3.1. The Case of Spain

In February 2020, the first cases of COVID-19 were detected in Spain. It is precisely because of the spread of the disease and the high degree of vulnerability of the population due to the high life expectancy that, from 14 March to 21 June, the Spanish government declared a state of alarm as the most restrictive measure to manage the health emergency through the approval of RD 463/2020. Despite this, after the end of this measure of social isolation, which had significant social and economic repercussions, there was a new resurgence of the virus, much more dizzying. In this sense, different organizations expressed their disagreement with the discredit of the notification system, with the non-existence of clear and agreed indicators, the insufficient capacity for early diagnosis and case follow-up, as well as the non-compliance with isolation measures and quarantine among the civilian population [19]. This epidemiological situation caused a methodological and didactic revolution because it meant that teachers had to put their knowledge, skills and digital skills into practice to face a great challenge. In this sense, many teacher educators thought of new ways to re-educate future teachers to provide them with strategies that allow them to face unpredictable and unknown scenarios from approaches to technological equity and learning [20].

### 3.2. The Case of the United States

The first identified cases of the COVID-19 disease in Colorado (USA) took place on 5 March 2020. In line with this and after the news broke that there could be 17 more suspected cases, on 10 March, the governor declared a state of emergency and also announced the opening of a medical center to carry out tests requested by doctors at no cost to patients. However, due to the low temperatures at the time, the center did not function as expected. To improve the situation and reduce the spread of the disease, on 18 March, the governor closed educational centers until 17 April.

## 4. Materials and Methods

### 4.1. Objective and Hypothesis

To answer the research question posed, the general objective of this study is to analyze the use that the active teachers made of fifteen groups of digital tools during the months of confinement caused by the COVID-19 pandemic in Spain and the United States as well as to examine the impact that this use had on their perceived self-efficacy, that is to say, on their beliefs regarding the implementation of these tools in the instructional process. Based on this objective, the following hypothesis is proposed:

**H1.** *The teachers’ perception of the use of digital tools is conditioned by socio–demographic variables such as geographic location, training received and the type of educational center*.

### 4.2. Participants

The participants in this study were 197 teachers (21.8% men and 78.2% women) of which 54.8% live in Spain and 45.2% live in the United States (see Figure 2).

Furthermore, as you can see in Table 1, regarding the age of the respondents, 21.8% are between 51 and 67 years old, 34.5% between 20 and 35 and 43.7% between 36 and 50. A total of 80.7% of those surveyed work in public centers, 12.2% in subsidized schools and 7.1% in private schools. Of these, 74.1% are career civil servants. On the other hand, 65% of the teachers surveyed stated that they teach in one educational level compared to 35% who state that they exercise their educational work in more than one educational level, in some cases reaching up to five. In line with this, 23.4% of the participating teachers work in the Primary Education stage, 15.2% in Vocational Training, 7.6% in Compulsory Secondary Education, another 7.6% in Baccalaureate, 6.6% in the University and 4.6% in Early Childhood Education. On the other hand, in relation to previous training in digital tools, 56.9% say they received it compared to 43.1% who did not.

### 4.3. Design

This study is based on a quantitative research design, supported by an exploratory descriptive and correlation approach in which the survey method is used through the design and application of an online questionnaire.

### 4.4. Questionnaire

EDU-COVID [21] is an online questionnaire (Appendix A) that consists of five sections: (i) informed consent, (ii) general data, (iii) use of fifteen web tools during confinement, (iv) assessment of use and (v) the effects on the teaching-learning process because of said use, for a total of 50 items. It is applied through the Google Forms web tool. This tool was chosen for various reasons, including cost-effectiveness since it is free, its accessibility, its simultaneity, its ease and its efficiency for the collection and extraction of data in multiple formats [21].

The Cronbach’s alpha coefficient is calculated to check the reliability of the instrument, a measure of internal consistency, which yields a value of α = 0.561, which indicates acceptable reliability. With the intention of evaluating the structure of the instrument based on the set of items that comprise it, a first Exploratory Factor Analysis was carried out. The suitability of the analysis was previously evaluated using the Kaiser–Meyer–Olkin (KMO) test and the Bartlett sphericity test [21].

The result of the KMO sample adequacy test is 0.617, with individual KMO measurements of the variables greater than 0.5. Furthermore, Bartlett’s test of sphericity was statistically significant (*p* < 0.001), indicating that the data are probably factorable. Therefore, these results reveal that the application of factor analysis is appropriate [21].

Next, the main factors were selected using the principal component extraction method to find a series of components that explain the maximum total variance of the original variables. Following Kaiser’s (1974) normalization rule, which establishes the extraction of principal factors from those with a p-value greater than one, nine components are obtained that explain 71.643% of the total variance, which can be considered a very acceptable value.

## 5. Procedure

To provide coherence and articulate the variables analyzed in the ad hoc questionnaire, a review of previously applied international instruments was carried out. After this, the online tool was designed using Google forms and sent to practicing teachers following the Delphi method to check the functionality, operability and to eliminate possible setbacks and difficulties that could be derived from the interpretation of the items.

Once the ambiguous items had been modified, they were sent to the teaching staff via email; social and academic networks; and the existing teaching groups on Facebook, LinkedIn and WhatsApp. The link was available for a month. At the end of the term, the resulting matrix was downloaded, the pertinent coding was carried out and the appropriate statistical analysis was also carried out using the SPSS program, version 28, which provided the empirical evidence of this study.

In line with the above, first, descriptive analysis related to the mean and the standard deviation was carried out to describe the participants. Then, we proceeded with the parametric analysis through the skewness and kurtosis tests that determine that the distribution complies with the normality assumption. Then, the reliability calculations of the instrument were made, including the internal item-scale consistency and Cronbach’s alphas. Finally, parametric analysis was performed through Student’s *t*-tests and ANOVA.

## 6. Results

### 6.1. Descriptive Analysis

First, the use and the beliefs of perceived self-efficacy by the teachers in different virtual teaching platforms were examined. For this, data about Google Classroom, Moodle, Classdojo, E-dixgal, Escholarium and others were collected. Of these tools, most teaching staff affirmed that they had not used E-dixgal (97.5%) and Escholarium (99%). In the case of the other options, it is striking that the participating teachers list digital tools for videoconferencing, such as Microsoft Teams, Zoom and Google Meet, as primary teaching platforms but not virtual learning environments such as Blackboard. Second, the descriptive statistical analysis was carried out to know the use of fifteen groups of digital tools by teachers, and the beliefs about their ability to use them effectively are described.

On the one hand, regarding the use of digital tools for instructional design, as can be seen in Figure 3, they are collaborative content editing applications such as documents, spreadsheets, Google or Microsoft or Prezi presentations, the most used by teachers during confinement. A total of 90.4% of the teachers surveyed affirm that they have used them and, of this group, 50.8% have used them daily. Followed by the use of interactive content creation tools such as Canva, Genially or Quizlet; online survey development through Google Forms or SurveyMonkey; audio and video recording tools such as Camstudio or Camtasia; and for the design of gamification such as Educaplay, Kahoot or Quizizz. Collaborative content editing websites such as blogs and wikis were the least used. In this sense, 57.9% of the teachers surveyed affirmed that they had not used blogs, and 79.7% wikis during confinement.

On the other hand, relative to the use of digital tools for communication and social interaction, as you can observe in Figure 4, the most used during the period of confinement by teaching faculty have been those for videoconferencing such as FaceTime, Skype and Google Meet; followed by those focused on the visualization and dissemination of videos (YouTube, Vimeo…); instant messaging (WhatsApp, Telegram, Bloomz) and social networks (Facebook, LinkedIn…). In this sense, 56.9% of the participating teachers affirm that they use videoconferencing tools daily, 38.6% instant messaging, 29.9% broadcast videos and 23.4% social networks. However, microblogging tools such as Twitter and Tumblr are shown as the least used preceded by applications for sharing photos and/or videos (such as Instagram, Flickr and Google Photos). Only 8.6% of those surveyed state that they use Twitter and Tumblr daily and 18.8% use the tools to share photos and/or videos.

### 6.2. Parametrics Analysis

A parametric analysis was also carried out and implies an estimate of the parameters of the population based on statistical samples, thus reducing the possibility of errors and increasing the degree of efficiency. In line with this, two types of tests are carried out: (i) the Student’s *t*-test for independent samples for the country of origin and the training received and (ii) the ANOVA for more than two independent samples for the type of educational center variables.

#### 6.2.1. Student’s *t*-Test for Independent Variables

As indicated above, through this test, the variable of the country of origin is analyzed, and statistically significant differences are obtained between populations in relation to the perceived use and self-efficacy of the virtual environments used and the fifteen digital tools selected. In this sense, differential patterns are observed in the analysis carried out of the perceived use and self-efficacy of teachers around the tools and applications examined according to the country of origin (see Table 2).

In Spain, the use of Moodle and Escholarium are the primary platforms for virtual teaching. In addition, the low use made during the confinement of wikis and programming tools is striking. It is also noteworthy that the abundant use of tools for videoconferencing versus other applications aimed at sharing images, communicating with friends, family and microblogging where less frequent use is evident. In contrast, the participants in the United States preferred to use Google Classroom over Moodle and Escholarium. Furthermore, participants in the U.S. also indicated more frequent use of cloud storage tools (Google Drive, Microsoft OneDrive, Dropbox), than participants in Spain. However, the use of instructional design tools like wikis and programming such as Joomla and Scratch remain incredibly low. On the other hand, the trend of use evidenced in Spain around digital tools for communication and social interaction is replicated in the United States; that is, videoconferencing tools were the most used, followed by social networks, image sharing sites and microblogging. In the case of the United States, however, the differences are more evident.

In view of the training received as a grouping variable, statistically significant differences are observed in the analysis of the use and perceived self-efficacy of the teachers around the tools and applications examined (see Table 3). In line with this, teachers who had received training in digital tools expressed greater perceived beliefs of self-efficacy around the use of digital tools in the teaching process, considering such use as more relevant. In addition, slightly higher scores are evident around the success of the teaching carried out during the confinement period both for the acquisition of competencies, as well as the achievement of objectives, the assimilation of the contents, and the obtaining of the learning results by those teachers who claim to have received training.

#### 6.2.2. ANOVA Test with More Than Two Independent Samples

As explained above, through this test, the type of educational center variables that are isolated as fixed factors are analyzed. In this sense, multivariate contrasts show significant differences when the kind of educational center is accounted [*λWilks* = 0.578, *F*_(328.000, 60.00)_ = 1.737; *p* = 0.002, *η*^2^ = 0.240] with medium-effect size.

As for the inter-subject effects, on one hand, considering the type of educational center as a grouping variable, statistically significant differences with medium and large effect sizes are shown, as can be seen in Table 4.

In relation to the post hoc, there is a pattern of differential use among teachers in public and concerted schools in favor of concerted schools and for the benefit of concerted teachers in multiple variables such as the use of Escholarium as a virtual teaching platform. [M_Public_ = 1 vs. M_Concerted_ = 1.08; *p* = 0.01]; the use of tools for video editing [M_Public_ = 2.61 vs. M_Concerted_ = 3.25; *p* = 0.01]; the use of digital tools or programming applications such as Joomla and Scratch [M_Public_ = 1.19 vs. M_Concerted_ = 1.92; *p* ≤ 0.01]; the use of digital tools to share videos and images [M_Public_ = 2.24 vs. M_Concerted_ = 3.13; *p* = 0.04] and the use of digital tools for microblogging [M_Public_ = 1.59 vs. M_Concerted_ = 2.50; *p* = 0.005].

## 7. Discussion and Conclusions

It should be noted that one of the key elements to ensure the success of educational digitization is teachers, as they are responsible for adapting and applying digital tools in the teaching and learning process. If teachers believe in the advantages and possibilities provided by technology and receive the necessary training and support, digital tools can be effectively integrated into the educational process. In this sense, the results described in the previous section show that the frequent and repeated use of digital tools by teachers does not imply that these applications are being successfully integrated into their teaching practices. In this sense, few teachers have sufficient understanding of the contribution of the fifteen digital toolkits examined and are therefore not benefiting from the potential of integrating them into the teaching and learning process.

At the same time, considering the functional characteristics of each of the tools analyzed, the results lead to a profile of the teacher more as a consumer of digital content than as the creator of digital content. However, it should be noted that a teacher with a high level of competence in the digital world should not only be able to use the technologies to enrich their teaching strategies, but also to propose and develop innovative practices based on the possibilities provided by the digital tools [22,23].

The results also show that the participating teachers do not feel fully prepared to integrate digital tools into the classroom [12,24,25,26,27]. In this sense, Tondeur et al. [27] in (2013) ensured that the training to integrate technology into the education that teachers receive is still in its beginnings. The educational restrictions during COVID-19 also created new barriers for teachers, and training would help them manage the added stress of using digital tools for remote instruction [28].

So, the Institutes for Teacher Training or Faculties and Schools of Magisterium are assigned the difficult and challenging task of providing education on digital integration [29]. Although the new generation of teachers is more digitally native than ever, mere technological skills and knowledge are not enough to integrate technology with guarantees of success [30] because this integration is a complex process in which many factors are involved, in addition to the digital competence level of the teacher.

Previous studies have shown that training in digital integration reduces anxiety about the use of technology [31,32] by providing pedagogical skills and technological experiences from specific content areas on where to apply digital tools [29,33]. In addition, teachers’ training and knowledge of various technologies are significant predictors of their use of technology in school [34,35]. Studies using the TPACK model also suggest that teachers’ attitudes and beliefs toward technology can have an impact on technology implementation [36]. Therefore, teacher training should not focus solely on how to use technology but should also emphasize how technology can improve the teaching and learning process.

All the issues mentioned above, including training and research on integrating digital tools into classrooms, must be addressed from multivariate approaches, understanding that educational digitization is the result of several personal, formative and contextual factors whose relationships can be very complex [37].

## Figures and Tables

**Figure 1 behavsci-13-00213-f001:**
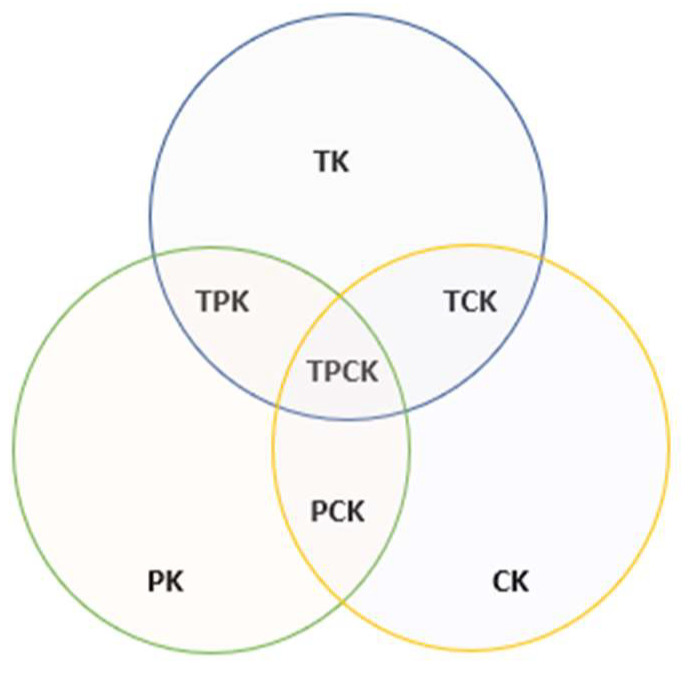
TPACK conceptual Model. Source: Taken from Koehler and Mishra, 2008 [9].

**Figure 2 behavsci-13-00213-f002:**
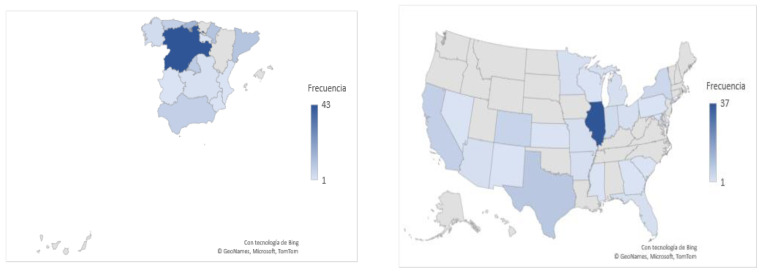
Distribution of the participants by autonomous community and state.

**Figure 3 behavsci-13-00213-f003:**
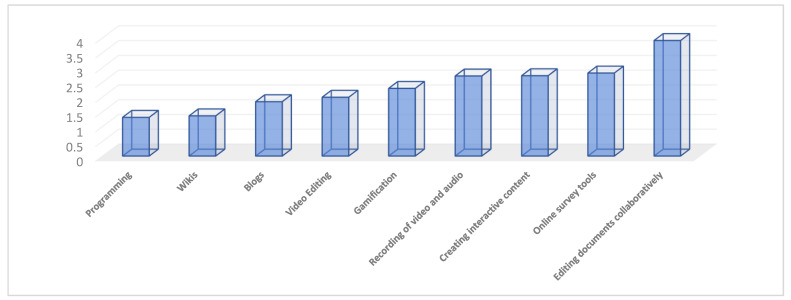
Average frequency of use of digital tools or applications for instructional design.

**Figure 4 behavsci-13-00213-f004:**
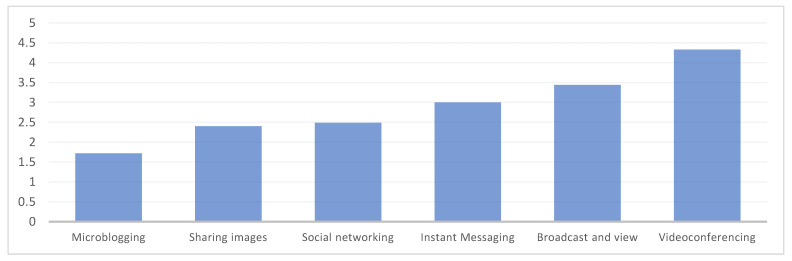
Average frequency of use of digital tools or applications for instructional design.

**Table 1 behavsci-13-00213-t001:** Sociodemographic data of the sample.

		Spain	United States	Totals
Gender	Men	34	9	43
	Women	74	80	154
Age	20–35 years	48	20	68
	36–50 years	44	42	86
	51–65 years	16	27	43
Type of Center	Public	75	84	159
	Charter (Private)	24	0	24
	Private	9	5	14
Career Officer	Yes	72	74	146
	No	36	15	51
Training Received on Digital Tools	Yes	55	57	112
	No	53	32	85

**Table 2 behavsci-13-00213-t002:** Student’s *t*-test accounting for the country of origin.

Variables on Perceived Use and Self-Efficacy	Spain	USA	Significance
Use of __________ as a primary platform for virtual instruction			
Moodle	2.34 (1.69)	1.15 (0.55)	≤0.001
Google Classroom	2.56 (1.82)	3.55 (1.61)	0.008
Escholarium	1.02 (0.13)	1.00 (0.00)	0.009
Tools for cloud storage (Drive, OneDrive, Dropbox…)	4.48 (1.04)	4.63 (1.04)	0.049
Use of digital tools or applications for the instructional design of			
Wikis	1.44 (0.93)	1.27 (0.70)	0.017
Programming (Joomla, Scratch…)	1.45 (0.97)	1.15 (0.51)	≤0.001
Use of digital tools for communication and social interaction applications...			
Videoconferencing	3.98 (1.05)	4.75 (0.62)	≤0.001
Sharing images	2.68 (1.67)	2.07 (1.51)	0.014
Social media interaction	2.82 (1.75)	2.08 (1.49)	≤0.001
Microblogging	1.96 (1.43)	1.43 (1.03)	≤0.001

**Table 3 behavsci-13-00213-t003:** Student’s *t*-test based on whether or not they received training.

Variables on Perceived Use and Self-Efficacy	No	Yes	Significance
Use of __________ as a primary platform for virtual instruction			
E-dixgal	1.01 (0.10)	1.05 (0.295)	0.012
Escholarium	1.00 (0.00)	1.02 (0.13)	0.013
Other	2.53 (1.75)	2.96 (1.88)	0.006
Cloud storage tools (Drive, OneDrive, Dropbox…)	4.67 (0.793)	4.46 (1.09)	0.005
Use of digital tools or applications for instructional design of...			
Recording video and audio	2.72 (1.41)	2.71 (1.59)	0.011
Programming	1.38 (0.91)	1.27 (0.72)	0.051
Watch and broadcast videos	3.31 (1.55)	3.54 (1.33)	0.007
Your use of these tools has been...			
Irrelevant versus relevant	4.58 (0.66)	4.72 (0.54)	0.003
The teaching carried out ensures that...			
Acquiring skills	1.38 (0.48)	1.46 (0.50)	0.033
Achieving the objectives	1.34 (0.47)	1.43 (0.49)	0.014
Assimilation of content	1.40 (0.49)	1.51 (0.50)	0.035
Obtaining the learning results	1.28 (0.45)	1.3 (0.48)	0.006

**Table 4 behavsci-13-00213-t004:** Testing of inter-subject effects based on the type of center.

	Public	Concerted	Private	F	*p*	*η* ^2^
Use of Escholarium as a primary institutional platform for virtual instruction	1.00 (0.00)	1.08 (0.28)	1.00 (0.00)	7.69	0.001	0.074
Using tools for video editing	2.61 (1.51)	3.25 (1.53)	2.93 (1.38)	5.43	0.005	0.053
Use of digital tools for instructional programming design	1.19 (0.66)	1.92 (1.17)	1.64 (1.00)	10.59	≤0.01	0.099
Using tools to share images and videos	2.24 (1.56)	3.13 (1.72)	3.07 (1.81)	4.45	0.013	0.044
Using microblogging tools	1.59 (1.20)	2.50 (1.69)	1.93 (1.14)	5.59	0.004	0.055
Non-functional versus functional	4.05 (0.88)	4.38 (0.82)	4.57 (0.64)	3.49	0.032	0.035

## Data Availability

Any information on the instrument or the original data will be provided by the authors if required.

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
