# Peer review of "The Perceived Self-Efficacy of Teachers in the Use of Digital Tools during the COVID-19 Pandemic: A Comparative Study between Spain and the United States"

_behavsci, 2023, doi:10.3390/bs13030213_

Round 1

Reviewer 1 Report

The information in Figure 3 is not visible, you need to improve it. Interesting study. Self-report regarding use and efficacy. The instruments are not clear, contribution in annexes to enable replication by other researchers. You should contribute many more recent international studies from impact journals. I have my doubts that the perceived usefulness can be measured as a quantitative variable, with the tests (ANOVA) you use. The intervention should be much more detailed. What use is made in practice of these tools, with details for the readers. More detail on reliability and validity.

You should improve discussion

Author Response

Dear reviewer:

We greatly appreciate your valuable feedback. These have made it possible to considerably improve the overall quality of the text.

In this sense, and taking into account their recommendations, figures 3 and 4 have been modified. We hope that the new format will be more legible.

Likewise, we have included in annexes the version of the questionnaire applied and information on its properties has been added.

Similarly, recent research on the subject has been included in the theoretical framework section.

And finally the discussion has been improved.

Reviewer 2 Report

In this paper, the author(s) examine the relationship between self-efficacy and the use of online/virtual instructional tools during the pandemic in the U.S. and Spain. Overall, the paper is clear and well-organized and appropriate for the journal. There are some strange grammatical phrasing in spots, but it is overall appropriately written. 

The biggest challenge for the paper is that theory is largely absent and, in this way, the analysis seems very ad hoc; it doesn't emerge from a theoretical angle that helps us understand how to use this information in the future. I also don't see the relevance of the COVID context in theorizing how these patterns might be different from other digital platform use. 

The paper references the TPACK model, but doesn't integrate it well into the manuscript. Instead of strongly-developed hypotheses which emerge from deep theoretical discussion, we instead only have garden-variety comparisons between what are more confounds than focal variables. 

As for self-efficacy, there is no attempt to logically connect the theory (Bandura) to the use of digital tools and, in particular, the tools studied. Why are the use of these tools purported to be related to self-efficacy?

What makes each of these very different digital tools unique? They are not the same nor do they have the same instructional uses/capabilities. I think it would have been better to focus on particular tools that carry out more or less the same functions. 

Then there is the sample and the COVID context. It is tricky to examine two different countries with very different responses to COVID overall. There is little discussion of this. Further, the pandemic lasted nearly two years and a lot changed in that time in both these countries. My recollection was that Spain was locked down during the study time, but the response in the U.S. was different and there was some variation in how localities responded to the pandemic. 

Other concerns about the study are the sample of teachers (very small sample relative to the sample frame of both countries), as well as the low reliability of the instrument. The author(s) report that reliability was adequate, but .561 is low. It is not clear what self-efficacy measure was used. Was the TSES (Teachers' Sense of Efficacy Scale; Tschannen-Moran and Woolfook-Hoy) used or something else? This is not clear. 

Overall, I would encourage the author(s) to devise a series of analyses that get at a more substantive question related to self-efficacy and digital tool use that has implications/application for the future. I see very little application/knowledge to be gained from the current findings, or perhaps the value of the findings is lost. Any pandemic study, in my opinion, needs to have as its central purpose, "What can we learn about what happened during this difficult time that will make virtual learning better next time?" I don't really see how the study helps us to answer this question. 

Author Response

We are very grateful to Reviewer 2 for his valuable comments, suggestions, and recommendations.

Throughout this time and in order to increase the quality of the manuscript we have tried to implement each one of them in the best possible way. In this sense, we hope that each of the changes made will be to your liking.

In line with the above, we have included an additional document in which we respond to each of the aspects discussed.

Round 2

Reviewer 1 Report

Ok, congratulations

Author Response

Thank you so much for your comment.

Reviewer 2 Report

I commend the authors for their thorough and careful attention to my comments. I do feel the paper is much improved as a result. The only issue that remains is that there are many grammatical issues that need attention in the manuscript, particularly in the highlighted parts that were added. The very first sentence of the abstract, for example, is a run-on sentence. Thanks! Tim

Author Response

Thank you so much for your suggestion and comment. We have improved the English in the manuscript.